



# Observations of Ionospheric Disturbances Associated with the 4 August 2020 Port Beirut Explosion by DMSP and Ionosondes

Rezy Pradipta[1,*] and Pei-Chen Lai[1,*]

[1]Boston College, Institute for Scientific Research, 140 Commonwealth Avenue, Chestnut Hill, MA 02467, United States
[*]These authors contributed equally to this work.
**Correspondence:** Rezy Pradipta (rezy.pradipta@bc.edu)

**Abstract.** A major explosion happened in Beirut on 4 August 2020, releasing a significant amount of energy into the atmosphere. The energy released may have reached the upper atmosphere and generated some traveling ionospheric disturbances (TIDs), which may affect radio wave propagation. In this study, we used data from the Defense Meteorological Satellite Program (DMSP) and ground-based ionosondes in the Mediterranean region to investigate the ionospheric response to this historic explosion event. Our DMSP data analysis revealed a noticeable increase in the ionospheric electron density near Beirut area following the explosion, accompanied by some wavelike disturbances. Some characteristic TID signatures were also identified in the shape of ionogram traces at several locations in the Mediterranean. This event occurred during a period of relatively quiet geomagnetic conditions, making the observed TIDs likely to originate from the Beirut explosion, and not from other sources such as auroral activities. These observational findings demonstrate that TIDs from the Beirut explosion were able to propagate over longer distance beyond the immediate areas of Lebanon and Israel/Palestine, reaching the Mediterranean and Eastern Europe.

## 1 Introduction

A chemical explosion of approximately 2.75 kilotons of ammonium nitrate occurred in a storage warehouse at Port Beirut at ~15:08 UTC on 4 August 2020 (Guglielmi, 2020; Pilger et al., 2021). This accidental man-made explosion had been estimated to have an effective yield of 0.8-1.1 kilotons of trinitrotoluene (TNT) to generate powerful shock waves in the atmosphere (Pilger et al., 2021), causing a significant property damage and tragic loss of lives. It was estimated that at least 135 people were killed and more than 5000 were injured (Singhvi et al., 2020). The estimated property damage, economic losses, and public reconstruction needs were between 8.5-10.1 billions USD (World Bank, 2020). The blast progression had also been analyzed using still frames and images from video surveillance footage overlooking the port (Diaz, 2021). With such approach, numerical fit for the size of the expanding fireball as a function of time provided an estimated energy yield of 0.6 kilotons of TNT equivalent. Meanwhile, analysis of videos that were posted on social media produced an estimate of explosion strength between 0.5-1.12 kilotons of TNT equivalent (Rigby et al., 2020). The scale of the Beirut explosion carried sufficient amount of energy which could have propagated through the upper atmosphere and affected the ionosphere, potentially generating some acoustic-gravity waves (AGWs) and traveling ionospheric disturbances (TIDs).





TIDs are wavelike disturbances that appear in the Earth's ionosphere in the form of periodic striations of electron and ion density (e.g. Hooke, 1968). Based on the scale size/wavelength of the striations, TIDs can be categorized into small-scale TIDs with wavelength of $< 100$ km, medium-scale TIDs (MSTIDs) with wavelength of 100-300 km, and large-scale TIDs (LSTIDs) with wavelength of $\sim$1000 km or larger (e.g. Hunsucker, 1982; Fedorenko et al., 2011; Ivanova et al., 2011 ; Habarulema et al., 2013; Boyde et al., 2022). TIDs can be a manifestation of AGWs at ionospheric/thermospheric altitudes, due

to coupling/collisions between neutral particles and charged plasma particles (e.g. Hines, 1960; Yigit and Medvedev, 2015). There exist various natural sources of AGWs and TIDs including solar flares (Zhang et al., 2019), geomagnetic storms (e.g. Nicolls et al., 2004; Pradipta et al., 2016; Zakharenkova et al., 2016; Jonah et al., 2018), volcano eruptions (e.g. Cheng and Huang, 1992; Themens et al., 2022; Takahashi et al., 2023), meteor explosions (e.g. Yang et al., 2014; Perevalova et al., 2015; Pradipta et al., 2015), and earthquakes/tsunamis (e.g. Tsugawa et al., 2011, Chou et al., 2020; Pradipta et al., 2023). Sometimes

there are also man-made sources of TIDs such as nuclear weapon detonation tests (e.g. Park et al., 2011; Huang et al., 2019), rocket launches (e.g. Chou et al., 2018; Liu et al., 2018), and major industrial accidents (e.g. Jones and Spracklen, 1974; Krasnov et al., 2003; Galushko et al., 2008). Depending on the scale of energy associated with a source, the distance traversed by the TIDs may vary, and the observed effects could have been short-lived and localized (e.g. Nishioka et al., 2013), or long-lived and global (e.g. Lee et al., 2008). Ionospheric plasma density fluctuations associated with TIDs can affect high-frequency

(HF) radio communications, navigation systems, and other technologies that depend on radio wave propagation properties of the ionosphere (e.g. Belehaki et al., 2020; Zhang et al., 2022).

Lower and upper atmospheric disturbances associated with the 4 August 2020 Beirut explosion had been investigated in a number of recent research works. The propagation of infrasonic waves near the Earth's surface from the Beirut explosion were characterized by Pilger et al. (2021) using the Comprehensive Nuclear Test Ban Treaty Organization (CTBTO) sensors.

It was estimated that the infrasonic waves propagated radially away from Beirut with a velocity of 344 m/s. Seismometers located in the region surrounding Beirut also indicated the presence of impulsive signals from the explosion in the form of seismic, hydroacoustic, and acoustic waves within 0.5-0.8 Hz frequency band (Pilger et al., 2021). The propagation of TIDs in the F-region ionosphere associated with the Beirut explosion had previously been reported by Kundu et al. (2021) and Jonah et al. (2021). In the two aforementioned studies, total electron content (TEC) data from ground-based global navigation

satellite system (GNSS) receivers in the Lebanon and Israel/Palestine area shortly after the explosion were analyzed, and TID propagation velocity in the 750-800 m/s range was found.

In this study, we investigated the TIDs associated with the Beirut explosion in more distant regions using in situ ion density measurements and ground-based ionosondes. This paper is organized as follows: in Section 2 we describe the methodology, in Section 3 we present the results and discussion, and in Section 4 we summarize the result and present the conclusion.



## 2 Data and Methodology

In this study, we used data from a variety of sources, including the Defense Meteorological Satellite Program (DMSP) and ground-based ionosondes in the Mediterranean region. Data on solar wind and geomagnetic conditions were also examined to assess the background space weather condition. Below we present the details.

### 2.1 DMSP

The DMSP satellites fly in low-earth orbit (LEO) at approximately 840 km altitude along sun-synchronous, near circular polar-orbits with an inclination of $98.7°$ (Burke et al., 2004). The orbital period is $\sim$104 minutes and it results in an averaged 14 orbits per day. The ascending nodes are in the dusk sector and descending nodes are in the dawn sector. For space weather study, the Special Sensor Ion Electron Scintillation (SSIES) instrument on board DMSP spacecrafts provides local information of number densities, temperatures, and drift motions of ionospheric ions and electrons. We specifically used 1-second Langmuir probe data from SSIES on DMSP F17, which provide measurements of heavy ($O^+$) as well as light ($He^+$, $H^+$) ion densities. The total ion density data are given with 1-second time resolution. On the other hand, the ion fraction data are originally given at 4-second resolution, and for our analysis they are interpolated onto the 1-second resolution time grid. We focused on the data from DMSP F17 based on the timing of its ascending pass over the Mediterranean area on 4 August 2020 following the explosion event at the Port Beirut. The ion densities were detrended using a 50-point moving average to reveal the presence of fluctuations indicative of ionospheric disturbances. The DMSP observation data were retrieved via the Madrigal website at http://cedar.openmadrigal.org.

Figure 1 shows a regional map containing the ascending orbit trajectories of the DMSP spacecrafts near the Beirut sector on 4 August 2020, along with the coordinates of a few key locations. The trajectories of different DMSP spacecrafts are indicated by different colors. At the two ends of each orbit trajectory line segment, the UTC values are shown. Given the time of the Beirut explosion (15:08 UTC), only the DMSP F17 spacecraft had the appropriate timing for its orbit to sample the ionosphere over Beirut's longitude sector shortly ($\sim$1 hour) after the explosion. Other DMSP spacecrafts passed over the area too early (before the explosion occurred) and their subsequent ascending passes after the explosion were too far ($\geq 20°$ lon) west of Beirut's longitude sector. The location of Port Beirut is indicated by a red asterisk, and the approximate positions of ionospheric disturbances intercepted by DMSP F17 along its trajectory (details presented in Section 3) are shown as red circles. Red squares mark the location of ionosondes, which we describe next.

### 2.2 Ionosondes

An ionosonde is a type of frequency-swept high-frequency (HF) radar that transmits radio waves vertically to measure various layers in the ionosphere. The observations from an ionosonde are typically displayed in the form of ionograms. The characteristic TID signatures in ionosonde observations may be recognized as periodic time variations in $f$oF2 or $h$mF2 (Klausner, 2009; Chernogor, 2015; Huang et al., 2016); or as special features in the shape of the traces in individual ionograms (Munro,



1950; Munro, 1953; and Heisler, 1958). These special ionograms trace features include: Y-forking of the trace near $f$oF2 , Z-twisting of the trace, additional cusp(s) in the trace, and formation of round loop(s) in O-mode and/or X-mode traces.

In this study we used three ionosondes: Nicosia (35.03°N, 33.16°E), Athens (38.00°N, 23.5°E), and San Vito (40.60°N, 17.80°E) stations, marked as red squares on the map in Figure 1. The Nicosia, Athens, and San Vito stations are approximately

250, 1172, and 1730 km distance from Beirut, respectively. At Nicosia and Athens, ionograms were recorded once every 5 minutes. Meanwhile, at San Vito, ionograms were recorded once every 7-8 minutes. We examined the time variations in the F2-layer virtual height $h'$F2 and critical frequency $f$oF2 to try to identify periodic wavelike behavior. In addition, we also examined the shape of the traces in individual ionograms to identify anomalous special features that would indicate the presence of TID passing over the ionosonde station. The ionogram data were retrieved from the University of Massachusetts Lowell (UML)

Global Ionospheric Radio Observatory (GIRO) Digital Ionogram Database (DIDBase). The data can be downloaded from the UML GIRO Data Center (https://giro.uml.edu) and the data can also be accessed using the SAO-Explorer program.

### 2.3   Solar Wind and Geomagnetic Activity Data

To determine external factors that could affect space weather conditions during the time period of interest, we examined a number of variables that are provided by the NASA OMNIWeb service (https://omniweb.gsfc.nasa.gov). For this study, we

retrieved the solar wind parameters which include the solar wind speed and the interplanetary magnetic field (IMF) components in GSM coordinates; geomagnetic indices which include the planetary-K (Kp) and the disturbance storm time (Dst) indices; as well as the F10.7 solar flux index. In addition, we also examined the solar flare index data (Dodson and Hedeman, 1975), which are prepared by the Kandilli Observatory and Earthquake Research Institute at the Bogazici University. The solar flare index data were retrieved from the NOAA National Geophysical Data Center (NGDC) at https://www.ngdc.noaa.gov/stp/

space-weather/solar-data/solar-features/solar-flares/index/.

### 3   Results and Discussion

Figure 2 shows a set of time series plots for basic solar wind parameters (IMF $B_z$ and solar wind speed $V_{\mathrm{SW}}$), geomagnetic indices (Kp and Dst), and F10.7 solar flux index on 3-5 August 2020. There was no significant southward turn of IMF $B_z$ and the solar wind speed was relatively steady between 400-600 km/s. Kp was slightly elevated on 3 August 2020, but it stayed

under $4^-$ and it dropped below $2^o$ at 06:00 UTC on 4 August 2020. Dst exhibited a slight activity reaching -30 nT on 3 August 2020, but it recovered quite quickly and it only hovered between 0 and -20 nT on 4-5 August 2020. The F10.7 solar flux index was stable at a relatively low value around 75 sfu. Examination of solar flare index data also indicated the absence of solar flares during this time period. In summary, solar and geomagnetic activity around the time of the Beirut explosion was rather calm, without any geomagnetic storm or solar flare. This quiet condition would help us in ruling out potential contamination

from space weather events.

Figure 3 shows a composite time series plot of ion density observations by the DMSP F17 spacecraft as it passed over Beirut's longitude sector between 16:00-16:30 UTC on 4 August 2020. Beirut's latitude is marked with a red arrow on the time





axis. Numerical values of some basic information (MLAT, LAT, LON, and MLT) of the DMSP F17 spacecraft orbit during this ascending pass are printed at the top portion of the graph. The bold blue (magenta) curves represent the light ions (oxygen ion)

density data recorded during this particular pass. The baseline trend lines (computed via moving averages) are superposed on top of these bold curves. The baseline for the light ions (oxygen ion) density is depicted as thin red (black) curve. Subtracting these baselines from the original observations gives us the net ion density fluctuations. The light ions (oxygen ion) density fluctuations are represented by the thin magenta (blue) curves. The time interval where significant level of fluctuations are present is marked with a pair of dashed lines, designated as $t_1$ and $t_2$. More precisely, the numerical boundaries of this time

interval are $t_1 = 16:22$ UTC $= 16.3650$ hr and $t_2 = 16:26$ UTC $= 16.4397$ hr. This highlighted interval corresponds to the red circles shown earlier on the regional map (cf. Figure 1). The peak-to-peak amplitude of these ion density perturbations is in the order of $4 - 5 \times 10^9$ m$^{-3}$.

Figure 4 shows a set of stacked line plots of oxygen ion density fluctuations recorded by the DMSP F17 spacecraft between geographic latitudes of $0°$N$-70°$N during ascending passes over Beirut's longitude sector at $\sim 18:00$ MLT for different calendar

dates from 4 June 2020 until 12 August 2020. The dates are selected such that all the specified ascending passes would be as close as possible to the pass happening on 4 August 2020 (i.e. the day of the Beirut explosion), with east-west spread of less than 500 km. As a reference, the day of the explosion is marked with a star sign. All of these DMSP F17 passes were at $\sim 18:00$ MLT sector, which would help us recognize if there were diurnally recurring TIDs near dusk for this longitude sector. In terms of the ion density fluctuation data, we found little or no sign of prominent TIDs that repeat diurnally around this area.

In addition, the potential effects of geomagnetic activity are taken into account using the Kp index. Two separate Kp index values are listed for each line plot: one is the maximum Kp over 24-hour period on that calendar date, and the other is the Kp value at 15:00 UTC on the same date. The specific hour of 15:00 UTC represents the approximate time-of-day when the explosion occurred on 4 August 2020. In these line plots, we found significant ion density fluctuations between geographic latitudes of $41°$N$-61°$N on 4 August 2020 (i.e. the day of the explosion); and practically none on the remaining calendar

dates. To help rule out possible contamination from geomagnetic activity, as Max Kp was $3^+$ on 4 August 2020, we compared these fluctuations against two other dates (4 and 5 July 2020) with similar Max Kp values of $3^o$ and $3^+$. We found little or no significant fluctuations on those two dates despite the slightly elevated Max Kp values. This pattern suggests that the ion density fluctuations intercepted by the DMSP F17 spacecraft over the aforementioned interval on 4 August 2020 was not due to geomagnetic activity. They could be related to the explosion at Port Beirut that occurred on that date approximately one hour

prior to the satellite pass over the area.

Figure 5 shows several representative ionograms from the Nicosia, Athens, and San Vito digisonde stations on 4 August 2020. The timestamp (in UTC) is indicated on the top left corner of each ionogram. Red/pink traces on these ionograms represent the O-mode echoes, and green traces represent the X-mode echoes. Traces with other colors represent O-mode echoes that arrive from oblique directions. For some ionograms, auxiliary panels are shown underneath to highlight certain

features (i.e. characteristic TID signatures) with magnification. Figure 5a shows a set of sequential ionograms recorded at the Nicosia station (codename NI135; 250 km away from Beirut) during 15:02-15:32 UTC interval. During this time interval, $f$oF2 was relatively stable around 5.5 MHz, without any sharp or drastic changes. However, 4 (four) ionograms are of interest to





us due to some anomalies in their trace shape between 3-5 MHz, which are highlighted with magnification in their auxiliary panels. At 15:12, 15:22, and 15:32 UTC, the O-mode trace can be seen forming a loop near 3 MHz. At 15:27 UTC, we can see

the O-mode trace forming a Z-shaped twist between 4-5 MHz. Figure 5b shows sequential ionograms recorded at the Athens station (codename AT138; 1172 km away from Beirut) during 15:15-15:45 UTC interval. During this time interval, there was a relatively thin (non-blanketing) sporadic-E layer, and we can see $f$oF2 varying slowly from 5.5 MHz to 5.0 MHz. While the $f$oF2 variation appeared smooth, 2 (two) ionograms are of interest to us due to some anomalies in their trace shape, which are highlighted with magnification in their auxiliary panels. At 15:25 UTC, the X-mode trace can be seen forming a loop. Later at

15:40 UTC, we can see the O-mode trace forming a loop. Figure 5c shows sequential ionograms recorded at San Vito station (codename VT139; 1731 km away from Beirut) during 15:30-16:15 UTC interval. During this time interval, there were more significant changes in the ionogram traces around $f$oF2. In fact, at 15:45 UTC the F2 trace momentarily disappeared. There was a strong sporadic-E layer (with some spread-E echoes) during this interval, which might have contributed to the aforementioned F2 trace disappearance (by blanketing). Further, 3 (three) ionograms are of interest to us due to some anomalies in their trace

shape, which are highlighted with magnification in their auxiliary panels. At 16:00 UTC, there was a Y-forking or bifurcation of the O-mode trace at $f$oF2. At 16:07 UTC, there was an upward-facing bulge in the O-mode trace just below 4 MHz. At 16:15 UTC, the aforementioned bulge evolved into a complete loop in the O-mode trace at approximately 3.5 MHz.

All of the above features in the ionogram traces (i.e. loop/mouth, Z-twist, and Y-forking) are indicative of TIDs propagating overhead (or near) the ionosonde stations (Munro, 1950; Munro, 1953; and Heisler, 1958). Such anomalies in the ionogram

traces arise from periodic undulations and curvatures in the isodensity contours of the F-region ionosphere due to AGW/TIDs (Munro, 1953; Cervera and Harris, 2013).

In addition to the ionogram trace anomaly analysis, we also examined time series data of auto-scaled $f$oF2 values from the GIRO DIDBase *FastChar* service for these 3 ionosonde stations — to look for ideal characteristic TID signatures in the form of ringing or sinusoidal signals. Unfortunately, we did not find any clear sign of long-lasting sinusoidal/oscillatory signals in

the $f$oF2 time series data that could be directly linked to AGW/TIDs from the Beirut explosion. The impulsive nature of the phenomenon may be the primary reason why the TIDs could be detected in the form of ionogram trace anomaly(ies), but not as long-lasting sinusoidal signals.

Given the recorded time of explosion at Port Beirut (15:08 UTC on 4 August 2020), we are able to determine the time delay since the explosion until the TIDs were either intercepted by DMSP or detected overhead by the ionosondes. Based on the

first arrivals of the TIDs, the shortest time delays since the explosion to each respective ionosonde station are: 4 minutes for Nicosia, 17 minutes for Athens, and 52 minutes for San Vito. In addition, great circle distances of the interception/detection points relative from Port Beirut is also known. Using both the time delay and distance information, we are able to assemble distance-time plot for these TIDs that we suspect to be caused by the explosion at Port Beirut. With the distance-time plot, kinematics analysis of the TID propagation is also possible.

Figure 6 shows a pair of distance-time plots for these TID interception/detection points, and the associated kinematics analysis of the TID propagation — with the explicit assumption that the TIDs had originated from Beirut. In each distance-time plot, great circle distances from port Beirut as a function of time delay since the explosion, based on DMSP and ionosonde



observations, are plotted. For comparison, auxiliary data points from previous works by Kundu et al. (2021) and Jonah et al. (2021) are also plotted. These auxiliary data points, shown in magenta and cyan, were obtained from a data figure in Kundu

et al. (2021) and from a data table in Jonah et al. (2021). Least-square fit lines to the DMSP and ionosonde data points using the model $y = m \cdot x$ (where $y$ is the distance and $x$ is the time delay) are displayed, with solid red line representing the main fit and dashed red lines representing the 95% confidence interval. In Figure 6a, data points that were used in the fit are shown as blue cross-hairs. For the ionosonde observations, this means all the data points. For the DMSP F17 observations, 3 (three) data points were included from the $t_1 - t_2$ time interval (cf. Figure 3): two at the interval boundaries and one at the midpoint.

Not all the DMSP observation data points were used so as not to overpower the ionosonde data points in the fitting process. In Figure 6b, data points that were used in the fit are shown as blue squares. These are the data points that represent the leading wavefront of the TIDs. For the ionosonde observations, these are the first ones to be detected since the explosion. Meanwhile for the DMSP F17 observations, it is the farthest data points from Beirut — i.e. data points intercepted at the $t_2$ epoch (cf. Figure 3). In both panels, data points that were not used in the fit are shown as light cross marks. The numerical fitting results

are shown on the top left corner of each panel. In the first treatment with roughly equal fitting weights for different observation types, shown in Figure 6a, the estimated TID propagation velocity was $492 \pm 94$ m/s. Here the main fit line falls almost exactly in between the measurement points from Kundu et al. (2021) and Jonah et al. (2021). The estimated TID velocity of $\sim$500 m/s, however, is well below the 750-800 m/s range that Kundu et al. (2021) and Jonah et al. (2021) had previously reported. In the second treatment with a priority for the leading wavefront, shown in Figure 6b, the estimated TID propagation velocity

was $649 \pm 193$ m/s. In this case, the main fit line is more closely aligned with the measurements from Jonah et al. (2021) as it corresponds to larger TID velocity that is generally closer to the aforementioned 750-800 m/s range, although one must note here that the statistical uncertainty is also larger.

From the data analysis, there are a few aspects of the present findings that we would like to highlight and elaborate further. These are: (1) the estimated TID propagation velocity that in general faster than the sound speed in air at ground level, (2) the

maximum radial range TID propagation away from the source at Port Beirut, (3) the absence of TIDs on the southern side of Beirut in the DMSP observation, and (4) the relative magnitude of the disturbances observed by different measurement types for this Beirut explosion event.

In comparison to regular sound speed of $\sim$330 m/s in air at ground level, the estimated TID propagation velocity in the range of 492-649 m/s from our kinematics analysis is significantly higher. The estimated TID propagation velocity of 750-800

m/s from previous studies based on GNSS TEC observations near Beirut area (Kundu et al., 2021; Jonah et al., 2021) is higher still. This faster-than-expected TID propagation velocity may be due to several factors. At ionospheric/thermospheric altitudes, the background temperature $T_n$ of neutral molecules is greater than that at ground level, which gives rise to a higher acoustic velocity $c_s \sim \sqrt{T_n}$ that would allow faster AGW/TID propagation velocity. Further, the explosive nature of the source may also contribute in causing atmospheric disturbances that propagate at supersonic speed, even at ground level. From controlled

explosive tests, it is known that the amount of overpressure determines the Mach number of the shock wave generated by the explosion (e.g. Kleine et al., 2003), with higher Mach numbers for greater overpressures. In the limit of vanishingly small overpressure, the Mach number would approach 1.00 (e.g. Medici and Waite, 2016). HF radio measurements of wave disturbances





in the lower ionosphere (i.e. at E-region altitudes) from explosive tests on the ground surface also yielded propagation velocity in the order of ∼1 km/s (e.g. Fitzgerald and Carlos, 1997). As such, TID propagation velocity in the range of 500-800 m/s

associated with the Beirut explosion event is quite reasonable. Our estimate for TID propagation velocity is lower than Kundu et al.'s (2021) and Jonah et al.'s (2021) estimates perhaps due to the fact that we are working with ionospheric disturbances detected at farther distances from Port Beirut.

In previous studies (Kundu et al., 2021; Jonah et al., 2021), the TIDs due to the Beirut explosion event were detected at distances ≤ 500 km from the source at Port Beirut. In those previous studies, the TIDs were observed in the TEC perturbation

(TECP) signals, where the disturbance signal from the Beirut explosion was recognizable in the form of an N-shaped impulse. Beyond the immediate areas of Lebanon and Israel/Palestine, no TID from the explosion event was detected in the TECP data. It is likely that the TID amplitude had dropped below the level that can be detectable in TECP data. Meanwhile in our analysis of DMSP and ionosonde observation data, the TIDs were detected/intercepted at distances up to 3000 km from Port Beirut, which are a considerably greater radial range. This greater range at which the TIDs can still be detected may come from a number of

factors. In the case of ionosondes, the detectability of the TIDs might be due to strong ionospheric refraction/bending of HF radio waves, making them highly sensitive to undulations that are present in isodensity contours of the bottomside ionosphere. This sensitivity would allow the ionosondes to be able to detect TIDs from the Beirut explosion up to a distance of ∼1700 km. In the case of DMSP, which is at an altitude of ∼840 km (i.e. topside ionosphere), the amplification of AGWs as they propagate to higher altitudes where the background atmospheric mass density is lower may help maintain the amplitude of the resulting

TIDs over large horizontal distance up to ∼3000 km. In turn, this amplification increases the chance of their detectability by DMSP at this topside ionospheric altitude.

In the DMSP F17 measurements during its ascending pass between 16:00-16:30 UTC on 4 August 2020, TIDs were intercepted between latitude $41°N-61°N$ to the north of Beirut. However, there was no corresponding plasma density disturbances observed to the south of Beirut. If everything is symmetrical, it is expected that TIDs from the Beirut explosion would be

detected in all directions. In fact, the TIDs reported in previous studies (Kundu et al., 2021; Jonah et al., 2021) were observed at point coordinates south of Beirut. There must be physical reasons that DMSP F17 did not intercept any TIDs to the south of Beirut at the altitude of its orbit. The reason might be related to the low ion-neutral collision frequency at the altitude of DMSP orbit, which makes the plasma behaves as magnetized plasma much more strongly than it does around the main ionospheric F-peak where ion-neutral collision frequency is higher. The ion-neutral collision frequency $\nu_{in}$ at 850 km altitude is

lower than that at 300 km altitude by a factor of $\sim 10^4$ (e.g. Tu et al., 2011). With magnetized plasma behavior, plasma density irregularities would be more forced to be field-aligned. AGW/TIDs that propagated southward from Beirut would arrive at low-latitude region where the geomagnetic field is nearly horizontal. As a result of this near-horizontal magnetic field and the magnetized plasma behavior at the DMSP altitude, the plasma density striations that formed would be field-aligned with horizontally elongated wavefronts. Undulations in plasma density would only be detectable along a vertical cut, not along a

horizontal cut — which makes it difficult for DMSP to detect. On the other hand, AGW/TIDs that propagated northward from Beirut would arrive at the upper midlatitude region where the magnetic dip angle is steeper. Under such geomagnetic field





geometry, the resulting field-aligned plasma density striations would still be visible along a horizontal cut — which makes it much more feasible for detection by DMSP.

Finally, we can make some remarks regarding the relative amplitudes of TIDs detected after the Beirut explosion event. In
Kundu et al.'s (2021) observations, the detected TIDs had a peak-to-peak amplitude of 0.28 TECU. The background TEC was 13.4 TECU, and thus the relative peak-to-peak amplitude was ∼2.1%. In Jonah et al.'s (2021) observations, the TID amplitude was 0.06 TECU (the peak-to-peak amplitude would be 0.12 TECU). The background TEC was approximately 11 TECU, and thus the relative peak-to-peak amplitude was ∼1.1%. In our analysis of the DMSP F17 observations, the peak-to-peak amplitude of the intercepted ionospheric disturbances was $4-5\times10^9$ m$^{-3}$ with a background ion density of $\sim 1.2\times10^{10}$ m$^{-3}$.
The relative peak-to-peak amplitude is therefore $3.3\%-4.2\%$. If we adopt Kundu et al.'s (2021) and Jonah et al.'s (2021) relative TECP amplitudes of $1.1\%-2.1\%$, then similar relative amplitude value can be expected for $f$oF2 fluctuations since a major contribution to TEC is the F-region ionosphere. With a background $f$oF2 value of ∼5 MHz, this implies a wave variation with amplitude of 50-100 kHz in $f$oF2. Given the frequency resolution of typical ionograms, $f$oF2 variations with this magnitude are difficult to detect — which might explain why we could not see any coherent oscillatory signals in the $f$oF2 time series
associated with this Beirut explosion event.

## 4 Conclusions

In this study, we analyzed the TIDs associated with the explosion that occurred at Port Beirut on 4 August 2020. Observation data from DMSP spacecraft and ionosondes in the Mediterranean region were used in the analysis. Quiet solar and geomagnetic condition on this date generally allowed us to rule out confounding factors from space weather activity. The DMSP F17
spacecraft intercepted ionospheric disturbances, in the form of ion density fluctuations, to the north of Beirut. The ionosondes detected ionospheric disturbances, in the form of anomalous ionogram trace shape (i.e. loop/mouth, Z-twist, and Y-forking), that propagated toward the west-northwest of Beirut. Two types of kinematics analysis based on the time-of-interceptions or time-of-arrivals and distance information yield TID propagation velocity estimates of $492\pm94$ m/s and $649\pm193$ m/s. These estimates were somewhat lower than the TID velocity estimates reported in two previous studies (Kundu et al., 2021; Jonah et
al., 2021) of 750-800 m/s. It was noted, however, that in the two previous studies the TIDs were observed at distances much closer ($\leq500$ km) to Beirut. Meanwhile, the TIDs analyzed in the present study were detected/intercepted at varying distances up to ∼3000 km from Beirut. Distinction between near-field and far-field properties of the wave disturbances might contribute to this difference. Overall, the present findings (based on DMSP and ionosonde data) show that the TIDs from the Beirut explosion were able to propagate over longer distance to reach the Mediterranean and Eastern Europe, beyond the immediate
areas of Lebanon and Israel/Palestine that were investigated (using GPS TECP data) in previous studies.

*Data availability.* DMSP observation data used in this study are available at the Madrigal database at **http://cedar.openmadrigal.org/**. Ionosonde data used in this study are available at the University of Massachusetts Lowell (UML) Global Ionospheric Radio Observatory



(GIRO) Digital Ionogram Database (DIDBase) at **https://giro.uml.edu/** or via the SAO–Explorer program. Solar wind parameter, solar flux, and geomagnetic index data are available from NASA OMNIWeb service at **https://omniweb.gsfc.nasa.gov**. The solar flare index data are

available from NOAA NGDC at **https://www.ngdc.noaa.gov/stp/space-weather/solar-data/solar-features/solar-flares/index/**.

*Author contributions.*    RZP conceptualized the study, analyzed the data, prepared the figures, and finalized the manuscript. PCL retrieved and analyzed the observation data, prepared first manuscript draft, and proofread the finalized manuscript.

*Competing interests.*    The authors declare that they have no conflict of interest.

*Acknowledgements.*    This work was supported by AFOSR grant FA9550-20-1-0313 at Boston College.



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



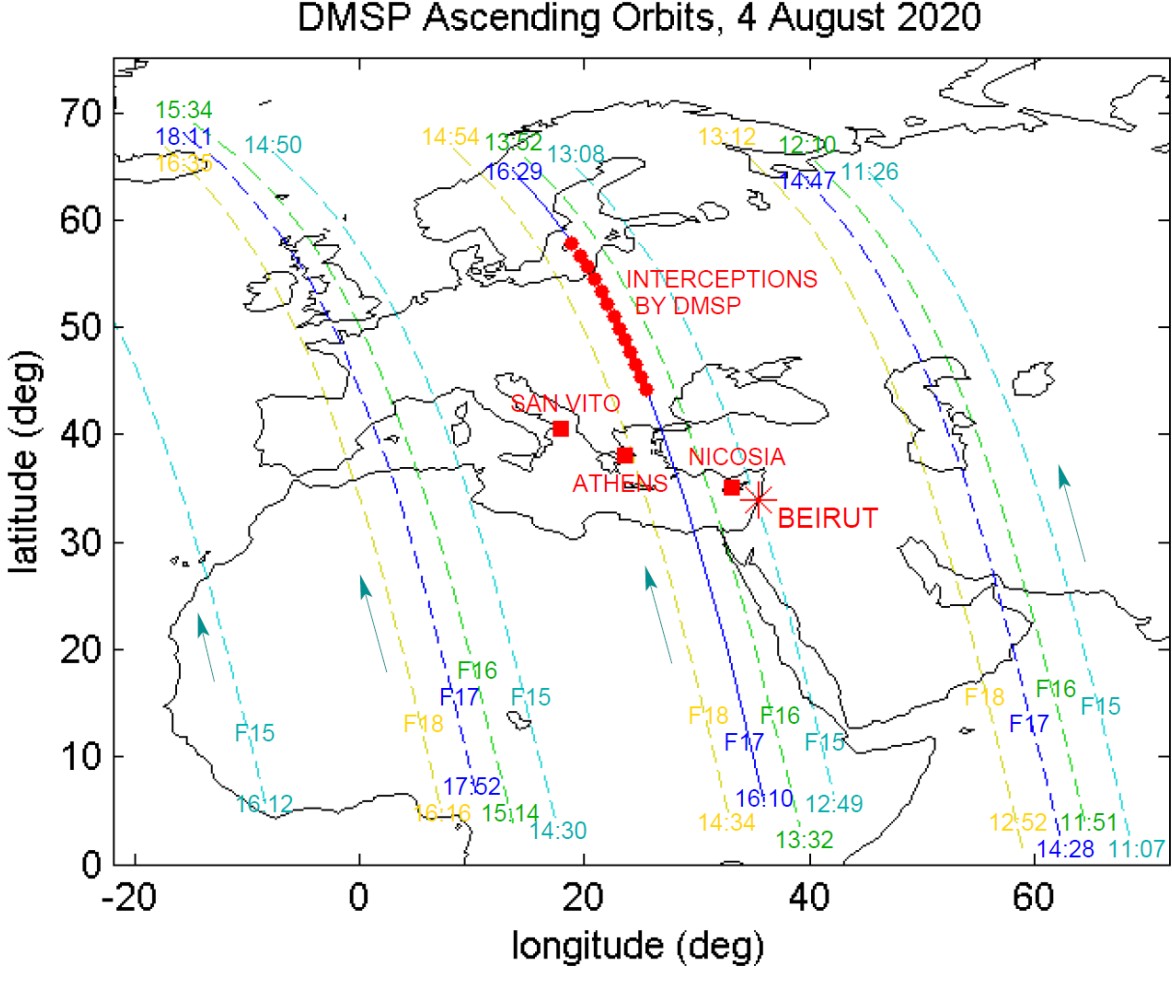

**Figure 1.** Situational map of the explosion site at Port Beirut, nearby ionosonde stations, and ground tracks of DMSP spacecrafts around the area on 4 August 2020. Arrows indicate the general orbit direction. The explosion occurred at 15:08 UTC.





**Figure 2.** Overview of solar and geomagnetic conditions on 3-5 August 2020. The time of the Beirut explosion (15:08 UTC on 4 August 2020) is indicated with vertical dashed red line.





**Figure 3.** Time series of ion density observations by DMSP F17 spacecraft between 16:00-16:30 UTC on 4 August 2020 over Beirut's longitude sector. The Beirut explosion occurred at 15:08 UTC, which was approximately 1 hour before this particular satellite pass.



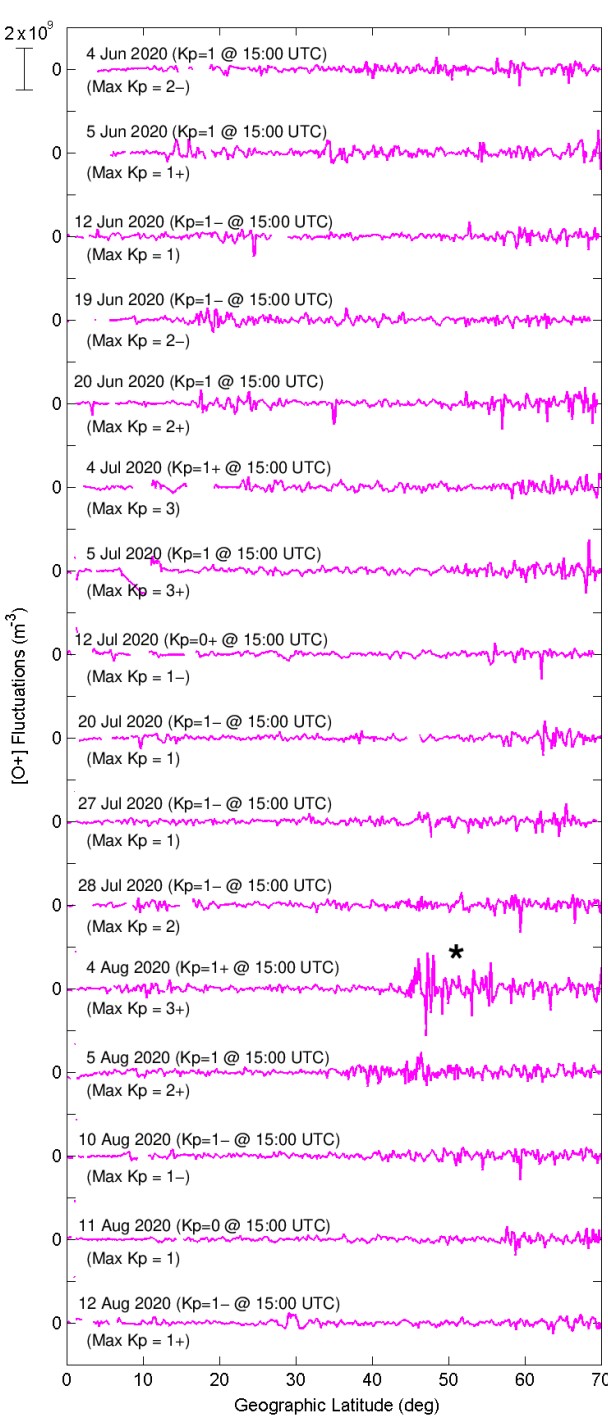

**Figure 4.** Oxygen ion density fluctuations registered by the DMSP F17 spacecraft between 0°N–70°N geographic latitude over Beirut's longitude sector at ~18:00 MLT for multiple dates before, on, and after the day of the explosion.



**Figure 5.** Sample ionograms from (a) Nicosia, (b) Athens, and (c) San Vito stations on 4 August 2020. Characteristic TID signatures, recognized as peculiar distortions in ionogram traces, are highlighted with magnification in auxiliary panels below the corresponding ionograms.



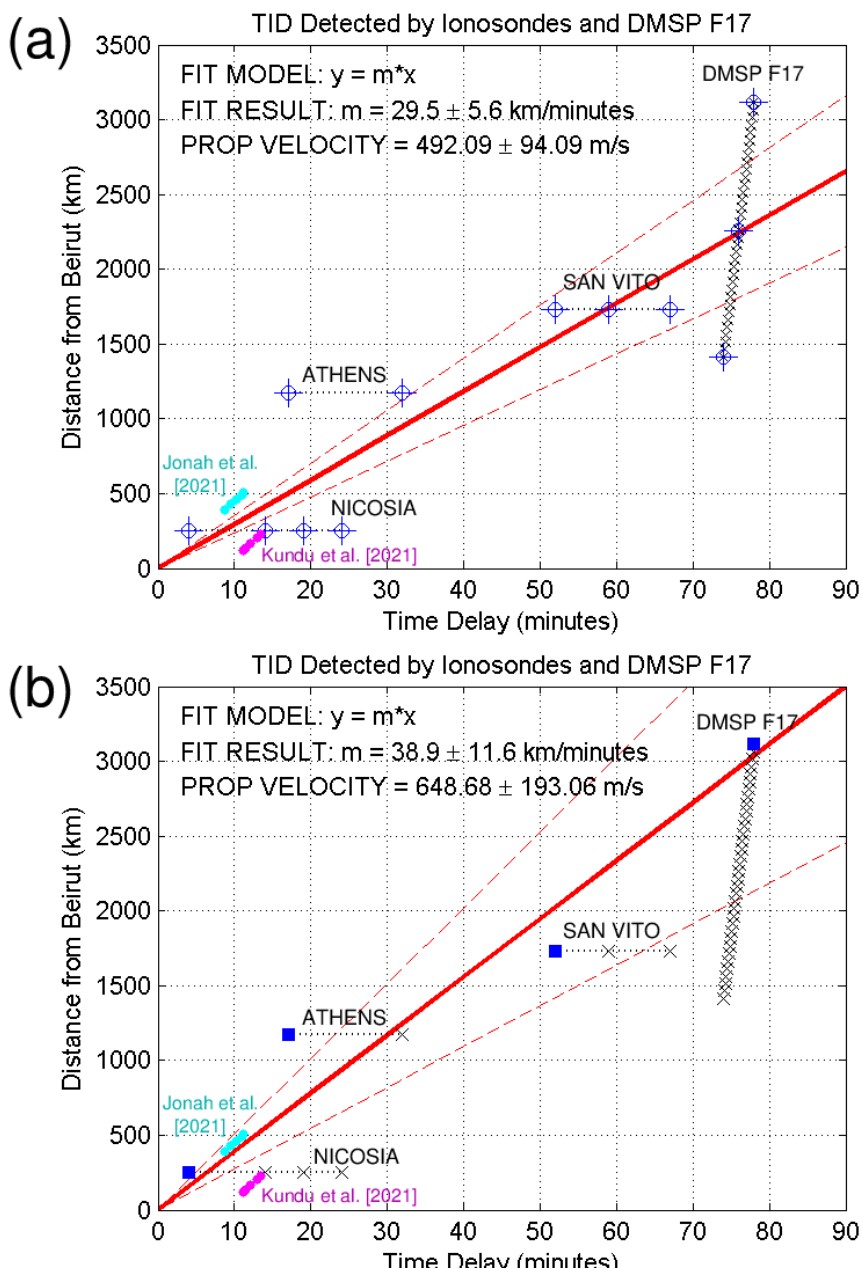

**Figure 6.** Kinematics analysis of TID propagation away from the explosion site at Port Beirut on 4 August 2020 (a) with equal weight applied to all types of data points in the curve fitting, and (b) with exclusive emphasis on data points associated with the leading wavefront in the curve fitting. Comparison with observations by Kundu et al. (2021) and Jonah et al. (2021) is included in the graphs.