# Peer review of "Observations of Ionospheric Disturbances Associated with the 4 August 2020 Port Beirut Explosion by DMSP and Ionosondes"

_Annales Geophysicae, 2023_

## Author Response (AR1)

**Response to Reviewers' Comments**

We would like to thank the editor and the referees for useful comments and questions on the material presented in the manuscript. Based on the referees' comments, we have made some modifications to the manuscript. In the annotated version of the revised manuscript (tracked changes), these modifications are highlighted in red colors. Below are our item-by-item response to the referees' comments. Here, our response is given in blue and/or red colors.

Response to Referee #1

The authors investigated TIDs as a result of Beirut explosion on 4 August 2020. To demonstrate that the resulting disturbances reached the ionosphere, they have used DMSP and ionosonde data. Characteristics found in ionograms at three different locations agree with earlier reported results pointing to AGWs/TIDs in the vicinity of the ionosondes. The study is carried out during a geomagnetically quiet period pointing to the explosion as the source of the observed wave-like structures. The authors have successfully showed increase in electron density around the Beirut area in DMSP data followed by what they have termed as TIDs seen in ionosonde data. This is a demonstration that the resulting TIDs reached the topside ionosphere. Previously, related studies were conducted using TEC data. The velocity values obtained in this study are lower than the earlier reported. The authors are requested to comment whether this was primarily related to energy dissipation during its evolution/propagation.

In my view, this paper is relevant and below are some minor comments which may be clarified before publication.

Around line 70, where the detrending was done using a moving average of 50 points. Please specify this in terms of time. For-example, if you detrend using a moving average of 60 minutes, you may miss some of the medium scale TIDs. Specifying time allows the reader to know what type of TIDs the analysis will reveal.

Regarding the 50-point moving average, this corresponds to temporal window of 50-second duration – given the 1-second time resolution of the DMSP ion density data. However, since the DMSP orbital speed is significantly faster than the TID propagation velocity (i.e. from the perspective of the DMSP spacecraft, it is as if the TIDs were almost stationary), assessing the situation in spatial dimension would be more meaningful. Orbiting at an altitude of ~840 km, DMSP has an orbital velocity of ~7.3 km/s. Hence, a 50-second time window corresponds to 50 sec × 7.3 km/s = 365 km spatial interval. The length of this spatial interval is greater than the conventional range for MSTID wavelengths (100-300 km), which means that we are in a comfortable zone for capturing MSTIDs with the 50-point moving average. Although, as the referee mentioned, some MSTIDs at the far corner (with wavelength of ~300 km) might be

somewhat suppressed. In the revised manuscript, we have now included some additional specifications (lines 70-74) in order to provide a clearer physical picture for the readers.

Figure 3, MLAT is combining with -31.8 making it positive. Create space between MLAT and -31.8

In order to avoid the merging between plot labels and numerical parameters, we will modify Figure 3 accordingly. In the revised manuscript, we have modified Figure 3 to create extra space between the MLAT label and its first numeric value, to fix the conjoining labels.

Lines 190-220; specify whether you are discussing meridional velocity or phase velocity of the associated TIDs in your comparisons with existing literature based on GNSS TEC analysis.

In this paper, we are characterizing the TID propagation based on their horizontal phase velocity away from the source region. This is the case for both the data we analyzed and the comparison with the existing literature. In the revised manuscript, we have now added a specification (lines 221-222) of this aspect in order to clarify it for the readers.

Lines 225-230 where you highlight that your velocities are lower than Jonah et al., (2021) and Kundu et al., (2021) because you are determining these values at further distances. Are you implying that the TID will have dissipated some energy? And based on this, how far do you think this TID will have travelled before the energy gets assimilated into background conditions?

Regarding the decay of the AGW/TID amplitude (and hence their detectability) at further distances, we believe that there are two factors in play: One is the spread of the AGW/TID wavefront over a wider area while conserving total energy; and the other is an actual energy dissipation. These are perhaps the main reason why TIDs from the 4 August 2020 Beirut explosion were not detected in the TECP measurements by Kundu et al. (2021) and Jonah et al. (2021) beyond the immediate area of Lebanon and Israel/Palestine regions. At further distances, we believe that the detectability of the TIDs from Beirut rests on the much higher sensitivity of HF diagnostics to bottomside ionospheric undulations (in the case of ionosonde observations) and amplitude amplification at the less dense topside ionospheric altitudes (in the case of DMSP observations). At San Vito ionosonde (1731 km from Beirut), we were still able to identify the characteristic TID signatures. Moving slightly further away, at Gibilmanna (1988 km from Beirut) and Rome (2199 km from Beirut) ionosondes, we were no longer able to clearly identify the TID signatures. Thus, if we have to speculate or make a prediction, we believe that the bottomside ionospheric undulations were dissipated at a distance of 2000-2200 km from Beirut. Meanwhile, for the electron density fluctuations at topside ionospheric altitudes, we believe that they would be dissipated as they crossed the auroral oval into the polar region. Given the solar local time sector of the DMSP F17 orbit

(~18:00 MLT) and Kp = 1+ condition, the auroral oval boundary can be estimated to be at 75° MLAT which was approximately 4800 km from Beirut. Thus, if we have to make a prediction, the electron density fluctuations at topside ionospheric altitudes probably did not survive beyond 4800 km distance from Beirut. In the revised manuscript, we have included some additional remarks on this matter (lines 186-191 and lines 263-266).

Response to Referee #2

This paper discusses the ionospheric disturbances associated with the man-made major explosion in Beirut on 4 August 2020. The authors used DMSP and ionosonde data to show the effect of the explosion in the upper atmosphere. They found an increase in ionospheric electron density near the Beirut area and showed some special features of ionogram traces associated with the TIDs. They concluded that the TIDs could propagate longer distances than previously reported.

Overall, this study is relevant and worth publication after a minor revision. Below are my comments:

1. Line 170 Cervera and Harris, 2013 should be Cervera and Harris, 2014

In the revised manuscript, we have now corrected this typographical error (line 176).

2. I think it is worth comparing the foF2 and h'F values on the explosion day with those on other quiet days. The passing TIDs might increase/decrease the foF2 and h'F values compared to the quiet days.

We performed an examination of foF2 and hmF2 variations with regard to this Port Beirut explosion event, as suggested by the referee. Here we used the auto-scaled foF2 and hmF2 from the UML GIRO FastChar for the 3 ionosonde stations (Nicosia, Athens, and San Vito). Based on the time series plots of foF2 and hmF2 alone (as well as the net ΔfoF2 and ΔhmF2 values), we could not find obvious/unambiguous signs of the TIDs. Rather, the characteristic TID signatures associated with the 4 August 2020 Beirut explosion had to be discerned based on the presence of anomalous traces in individual ionograms (i.e. loop/twist and Y-forking features). Nevertheless, we believe that the time series plots of foF2 and hmF2 (as well as ΔfoF2 and ΔhmF2) are still valuable for the readers. We have put these time series plots in the Supplementary Material, and we have also included some remarks on this matter in the revised manuscript (lines 193-198).

3. Could you please explain what's the mechanism between the forming of the loop in the bottom side of the F-layer and the Y-forking feature near the foF2? And why did only San Vito Ionosonde observe the Y-forking feature?

Regarding the formations of the characteristic loop/twist and Y-forking features seen in the ionograms, their underlying mechanisms can be understood based on Munro's (1953) and Heisler's (1958) explanations, which we may summarize here. These features are the direct result of a concave indentation in the ionospheric isodensity contour overhead the ionosonde station. This situation could materialize the when the ionosonde is located under a trough of the TIDs, with two opposing TID crests on either sides. This setting can be illustrated using the diagram below, borrowed from Munro (1953) with some modifications:

[Figure]

In Munro's (1953) paper, the indentation in the isodensity contour was constructed using a set of circular arcs and straight-line segments. The middle part of the indentation may be called the canopy/roof (in Munro's paper this was referred to as $Y3$). The two sides of the indentation may be called the brims (in Munro's paper these were referred to as $Y1$ and $Y2$). The brims smoothly extend to the ambient parts of the isodensity contour (represented here by the straight-line segments). Depending on the position of the ionosonde (at ground level) relative to the indentation, return signals may come from the left brim, the right brim, and/or the canopy. The special geometrical construction of the isodensity contour based on circular arcs and straight-line segments allows a straightforward rule for determining whether or not the ionosonde will receive a return signal in this simple mirror reflection model. A normal incidence on the isodensity contour would result in a return signal, where normal direction is pointing radially away from the center of the corresponding circle. Otherwise, there would be no return signal.

The loop/twist feature can occur if the ionosonde position is non-symmetrical with respect to the center of the indentation, and the TID wavefront is tilted at an angle from the vertical. On

the other hand, the Y-forking feature will occur if the ionosonde position is symmetrical with respect to the center of the indentation, and the TID wavefront has little/no tilt (i.e. relatively close to a vertical orientation). This geometrical configuration is illustrated in the following diagrams, adapted from Munro (1953) with some modifications.

[Figure]

We show five representative ionospheric isodensity contours at increasing heights, numbered accordingly. Sample rays from the ionosonde are drawn, generally aimed at either the brims or the canopy of the indentation at each contour level. Rays that are launched toward the left (right) brim are displayed with green (red) color; whereas rays that are launched toward the canopy are displayed with blue color. Some of these sample rays do not result in a successful return signal for the ionosonde, representing situations where normal incidence is impossible given the configuration. For clarity, if a ray results in a successful return signal, the location of the reflection is marked/highlighted on the corresponding isodensity contour line.

The diagrams on the left illustrate the configuration that gives rise to the loop/twist feature in an ionogram. Due to the tilt angle of the TID structure, in this case direct returns from the right brim (marked with red color) can only occur from the lowest set of isodensity contour lines (No. 1-3). At the higher contour lines (No. 4 and 5), the reflected rays cannot return because the right brim of the indentations is located too far toward the right to allow normal incidence. This causes the echoes from the right brim (labeled $Y1$) to appear only at the lower frequencies of the synthetic ionogram shown at the bottom. The opposite happens with the rays that are directed toward the left brim (marked with green color). In this case, due to

the tilt angle of the TID structure, direct reflections cannot happen at the lowest contour lines (No. 1) as the left brim is located too far toward the left. Direct returns from the left brim can only happen at the higher contour lines (No. 2-5) as the overhead contour straightens out at higher altitudes. This causes the echoes from the left brim (labeled $Y2$) to occur primarily at the higher frequencies in the synthetic ionogram. Meanwhile, direct returns from the canopy of the indentation (marked with blue color) can only happen at some intermediate altitudes (contours No. 2 and 3) due to highly selective focusing properties of a concave mirror. This causes the echoes from the canopy (labeled $Y3$) to appear only within a narrow frequency interval in the middle. Conceptually, such a clustered/uneven separation of the three types of rays (the canopy and left/right brims) at different altitudes creates the loop/twist feature in the synthetic ionogram.

The diagrams on the right illustrate the configuration that gives rise to the Y-forking feature in an ionogram. In this case, the ionosonde is simultaneously looking at two different cross-sections of the ionosphere: one directly overhead (reflection from the roof of the indentation) and the other oblique from the sides (reflections from the two brims of the indentation which are symmetrically positioned). Conceptually, this gives rise to the Y-forking feature in the synthetic ionogram.

This explanation, based on Munro (1953) and Heisler (1958), only involves a simple mirror model, which ignores ray bending. However, conceptually this simple mirror model already captured most of the basic essence and it provides good analytical intuition.

Regarding why the Y-forking feature was only observed over San Vito (but not over Nicosia and Athens) after the 4 August 2020 Beirut explosion, our lines of thinking are as follows. We believe that there are two potential factors: the strength of TID undulations at F1-layer height and the tilt angle of the TID wavefront. It is possible that when reaching San Vito, the strength of TID undulations at F1-layer (or E-F valley region) height had weakened. Moving further away from the explosion source, the local AGW/TID phase velocity might also have leveled out to become more horizontal, and the TID wavefront lost some of its tilt angle as a result. Indentations around F1-layer height and tilt angle in the TID wavefront are important ingredients for the formation of loop/twist features in the ionogram trace. As such, these two factors could have caused the change from loop/twist features (seen over Nicosia and Athens) into Y-forking feature (seen over San Vito).

In the revised manuscript, we have included some additional explanations on the mechanisms behind the loop/twist and Y-forking features in the ionograms, and plausible reasons on why the Y-forking feature was seen only at San Vito (lines 176-185). We have also included the above two schematic diagrams (and extended explanations) in the Supplementary Material.

4. Figure 1. Each symbol is described in the text but I think it is also a good idea to include it in the figure so it is easier for readers to understand the figure at a glance.

In the revised manuscript, we have added brief description(s) of the symbols/ parameters in Figure 1 for quick reference by the readers — in the figure caption.

---

## Author Response (AR2)

**Response to Reviewers' Comments**

We would like to thank the editor and the referees for their comments and questions. The finalized version of the manuscript has now been completed. In the annotated version of the manuscript (tracked changes), the modification history is highlighted in red colors.